# Hypoxia Differently Affects TGF-β2-Induced Epithelial Mesenchymal Transitions in the 2D and 3D Culture of the Human Retinal Pigment Epithelium Cells

**DOI:** 10.3390/ijms23105473

**Published:** 2022-05-13

**Authors:** Soma Suzuki, Tatsuya Sato, Megumi Watanabe, Megumi Higashide, Yuri Tsugeno, Araya Umetsu, Masato Furuhashi, Yosuke Ida, Fumihito Hikage, Hiroshi Ohguro

**Affiliations:** 1Department of Ophthalmology, School of Medicine, Sapporo Medical University, S1W17, Chuo-ku, Sapporo 060-8556, Japan; ophthalsoma@sapmed.ac.jp (S.S.); watanabe@sapmed.ac.jp (M.W.); megumi.h@sapmed.ac.jp (M.H.); yuri.tsugeno@gmail.com (Y.T.); araya.umetsu@sapmed.ac.jp (A.U.); funky.sonic@gmail.com (Y.I.); fuhika@gmail.com (F.H.); 2Department of Cardiovascular, Renal and Metabolic Medicine, Sapporo Medical University, S1W17, Chuo-ku, Sapporo 060-8556, Japan; satatsu.bear@gmail.com (T.S.); furuhasi@sapmed.ac.jp (M.F.); 3Department of Cellular Physiology and Signal Transduction, Sapporo Medical University, S1W17, Chuo-ku, Sapporo 060-8556, Japan

**Keywords:** TGF-β2, human retinal pigment epithelium, 3D culture

## Abstract

The hypoxia associated with the transforming growth factor-β2 (TGF-β2)-induced epithelial mesenchymal transition (EMT) of human retinal pigment epithelium (HRPE) cells is well recognized as the essential underlying mechanism responsible for the development of proliferative retinal diseases. In vitro, three-dimensional (3D) models associated with spontaneous O_2_ gradients can be used to recapitulate the pathological levels of hypoxia to study the effect of hypoxia on the TGF-β2-induced EMT of HRPE cells in detail, we used two-dimensional-(2D) and 3D-cultured HRPE cells. TGF-β2 and hypoxia significantly and synergistically increased the barrier function of the 2D HRPE monolayers, as evidenced by TEER measurements, the downsizing and stiffening of the 3D HRPE spheroids and the mRNA expression of most of the ECM proteins. A real-time metabolic analysis indicated that TGF-β2 caused a decrease in the maximal capacity of mitochondrial oxidative phosphorylation in the 2D HRPE cells, whereas, in the case of 3D HRPE spheroids, TGF-β2 increased proton leakage. The findings reported herein indicate that the TGF-β2-induced EMT of both the 2D and 3D cultured HRPE cells were greatly modified by hypoxia, but during these EMT processes, the metabolic plasticity was different between 2D and 3D HRPE cells, suggesting that the mechanisms responsible for the EMT of the HRPE cells may be variable during their spatial spreading.

## 1. Introduction

Human retinal pigment epithelial (RPE) cells, which are located between the neural retina and the choriocapillaris, form the outer blood–retina barrier (BRB), thereby having physiological roles in the maintenance of homeostasis of the retina and choroid [1]. Alternatively, they also appear to have pathological roles in retinal and choroidal neovascularization as well as retinal fibrotic diseases such as proliferative vitreoretinopathy (PVR) and age-related macular degeneration (ARMD) [2]. Among such pathogenic conditions, the epithelial mesenchymal transition (EMT) of the RPE is also recognized as a key underpinning mechanism and could potently be induced by transforming growth factor-beta 2 (TGF-β2). In fact, elevated TGF-β2 levels have been reported in the vitreous fluid of patients with PVR [3,4,5,6] and all of the mammalian TGF-β isoforms as well as TGF-β receptor type II have been detected in surgically obtained epiretinal membranes obtained from patients with PVR [7,8,9,10]. In addition, the EMT of RPE causes the BRB to be breached, thus facilitating the development of choroidal neovascularization (CNV) [10]. Regarding the development of CNV, it is also well known that hypoxia is also pivotally involved [11]. In fact, hypoxia is associated with the development of many diseases in the retina, including ARMD, diabetic retinopathy (DR), ischemic type of retinal-vein occlusion (RVO) and retinopathy of prematurity (ROP) [12,13,14] by stimulating the secretion of several proangiogenic cytokines, including vascular endothelial growth factor (VEGF) and erythropoietin (EPO), through a hypoxia inducible factors (HIFs) dependent mechanism. Among the various HIFs, HIF-1 is a master oxygen sensor as well as a major transcription factor that is specifically activated during hypoxia [15]. Therefore, as shown above, both the EMT of RPE cells and hypoxia are extremely important in terms of understanding the possible molecular mechanisms responsible for these retinal diseases, and both mechanisms may be closely linked. In fact, quite recently, Shu et al. reported that the TGF-β2-induced EMT of RPE-induced metabolic dysfunctions in mitochondria [16].

During the pathological progression of EMR in RPE cells, the resulting RPE fibrosis associated with neovascular vessels spatially spread toward both the neural retina and choroid. For example, in the case of the neovascular ARMD (neovARMD), neovessels sprout in either the choroidal or the subretinal directions to form choroidal neovascular membranes (CNV) or intraretinal angiomatous proliferations within the macular region [17]. The results of histological studies indicate that several cells including RPE, vascular endothelial cells, macrophages, pericytes, fibroblastic cells and myofibroblasts in addition to ECM proteins are present within the CNV membranes obtained from patients with neovARMD [18,19,20]. Furthermore, during the progression of this disease, it was histologically observed that degenerating RPE cells were reversibly transformed into mesenchymal cells via EMT to adapt such a harsh microenvironment [21,22,23,24]. Therefore, to understand the molecular mechanisms responsible for the spatially growing EMT changes in RPE cells, it becomes necessary to study the effects of both TGF-β2 and hypoxia, as possible major pathogenic inducers using a reliable 3D cell culture system that replicates such a spatial environment. In fact, in addition to the conventional 2D cell cultures, such an in vitro 3D spheroid model has been identified as a powerful tool for studying the molecular mechanisms of EMT as well as hypoxia within cancerous cells [25,26,27,28].

Therefore, to examine these issues in more detail, the TGF-β2-induced EMT of HRPE cells under normoxia and hypoxia conditions were subjected to the following analyses, employing our recently developed 3D cell cultures [29,30,31,32], in addition to conventional 2D cultures: (1) the barrier functions of 2D HRPE monolayers, (2) cellular mitochondrial and glycolytic functions (2D), physical properties of the 3D spheroids, and (3) the expression of ECM proteins, HIF1α and 2α, with the possible up-stream regulators requiring the generation of the 3D spheroids including *STAT3*, *IL6*, *FOS*, *TGFb1*, *AGT* and *MYC*, and a master regulator of mitochondrial functions, *PGC1a* (2D and 3D).

## 2. Results

To study the effects of hypoxia on the TGF-β2-induced epithelial-mesenchymal transition (EMT) of the retinal pigment epithelium (RPE), 2D and 3D cultured HRPE cells were prepared in the absence and presence of a 5 ng/mL solution of TGF-β2 under normoxia and under hypoxia conditions. Prior to the current analytical experiments, a cell viability assay confirmed that these experimental conditions were not toxic (Figure 1A). To evaluate the function of the putative outer blood retinal barrier (oBRB) that is comprised of RPE cells, we used the 2D HRPE monolayers as a simple in vitro model of oBRB. Within the 2D HRPE monolayers, the TGF-β2 treatment and hypoxia resulted in a significant increase in the TEER values of those untreated under normoxia conditions, and both types of stimulations synergistically induced further enhancement effects (Figure 1B). Similar to this, a mono-treatment with TGF-β2 or hypoxia also caused a significant decrease and increase in the mean sizes and stiffness of the 3D HRPE spheroids, respectively, and such effects were also synergistically enhanced by the simultaneous treatments of both stimulants (Figure 2).

To study this issue further, the expressions of major ECM proteins, including collagen1 (COL1), COL4, COL6, and fibronectin (FN) under the above conditions were examined by qPCR analysis by immunohistochemistry. An analysis by qPCR (Figure 3) indicated that, in the 2D HRPE cells, (1) *COL1*, *COL4* and *FN* were significantly up-regulated by TGF-β2 under both normoxia and hypoxia conditions, (2) the mRNA expressions of *COL6* and *FN* in the absence of TGF-β2 under hypoxia conditions were also significantly higher than those under normoxia conditions, and (3) the production of COL6 was substantially down-regulated by TGF-β2 under hypoxia conditions. Similar but lesser differences in the expression of ECM molecules were also observed among these experimental conditions as evidenced by immunohistochemistry (Figure 4). Alternatively, in the 3D HRPE spheroids, the mRNA expression of *COL1* and *FN* were also significantly up-regulated by TGF-β2 under both normoxia and hypoxia conditions, and the expression of COL4 and FN under hypoxia conditions were markedly higher than those under normoxia conditions (Figure 5). While, in contrast, the immunolabeling of these ECMs in the case of 3D HRPE spheroids was different compared to their mRNA expressions (Figure 6), that is, (1) TGF-β2-induced down-regulation of COL4 and COL6 or COL4 were recognized under both normoxia and hypoxia conditions, respectively, (2) in the absence of TGF-β2, the expression of COL1 under hypoxia conditions was significantly lower than that under normoxia conditions, and (3) in the presence of TGF-β2, the expression of COL4 under hypoxia conditions was significantly higher than those under normoxia conditions. The diversity between qPCR and the immunolabeling of the 3D spheroids was also reported in our previous study using cells from other sources [30,31,33,34,35]. As a possible explanation for this, we speculate that immunolabeling adequately reflects the expression of target molecules that are located on the surface of the 3D spheroids, while, in contrast, the qPCR analysis reflected the total expression. These collective observations indicate that EMT were significantly and synergistically induced by the TGF-β2 treatment and hypoxia in both 2D HRPE cells and the 3D HRPE spheroids.

Next, to elucidate possible underlying mechanisms responsible for causing such TGF-β2- or hypoxia-induced effects, the real-time cellular respiratory metabolic analysis of 2D and 3D HRPE cells was conducted in both the absence and presence of TGF-β2. That is, the oxygen consumption rate (OCR) and the extracellular acidification rate (ECAR) were simultaneously measured in real-time to assess mitochondrial respiration and glycolytic ability using a Seahorse XFe96 Bioanalyzer. (Agilent Technologies, Santa Clara, CA, USA) As shown in Figure 7A,B, the OCR after injecting the uncoupler FCCP, which reflects mitochondrial maximal respiratory capacity, was significantly decreased in the presence of TGF-β2 in the case of the 2D HRPE cells. Although such TGF-β2-induced reduction in mitochondrial maximal respiratory capacity was not observed in the 3D HRPE spheroids, the OCR after the injection of the ATP synthase inhibitor oligomycin, which reflects proton leakage, was increased in the presence of TGF-β2 in the 3D HRPE spheroids. In contrast, the difference between the basal ECAR and the ECAR values after oligomycin injection, which reflects glycolytic reserve, was significantly decreased in the presence of TGF-β2 in the case of the 2D HRPE cells, but this was not observed in the 3D HRPE spheroids. These results rationally suggest that (1) the TGF-β2-induced deterioration of OCR and ECAR was more evident in the 2D HRPE cultures rather than the 3D HRPE spheroid cultures, (2) TGF-β2-induced effects during the EMT process were similar with the hypoxia induced effects, and (3) some factors related to glycolysis and/or mitochondria oxidation may be involved in these mechanisms. To address this issue, the mRNA expressions of several possible candidate factors including hypoxia-related factors; *HIF1α* and *2α*, and a master regulator of mitochondrial functions, *PGC1a* were evaluated (Figure 8). As quite similar to the above physical properties of TEER of the 2D HRPE monolayer, and size and stiffness of 3D HRPE spheroids, the gene expression of HIF1α were significantly and synergistically up-regulated by stimulation with both TGF-β2 and hypoxia. While in contrast, mRNA expression of HIF2α was substantially down-regulated by TGF-β2 under normoxia or by a hypoxia stimulation in the absence of TGF-β2 in 2D HRPE cells, but no significant changes were observed among these four conditions in the 3D HRPE spheroids. In addition, those HIF1α and 2α expressions were significantly higher in 2D HRPE cells than in the 3D HRPE spheroids. These data rationally supported that HIF1α may be a key factor regulating the mitochondrial respiration and glycolytic abilities in response to TGF-β2 and/or hypoxia stimulations toward 2D and 3D HRPE cells as above. To support this idea, mRNA expressions of *PGC1a* were significantly elevated in the 3D HRPE spheroid as compared to the 2D HRPE cells, and those were also increased under hypoxia conditions (Figure 8).

To study further in terms of the difference between 2D and 3D HRPE cells under several conditions as above, qPCR analysis of the recently determined possible up-stream regulators forming 3D spheroids; *STAT3*, *IL6*, *FOS*, *TGFb1*, *AGT* and *MYC* using 3T3-L1 cells [36] were studied. As shown in Figure 9, among these regulators, mRNA expressions of *STAT3*, *IL6*, *FOS* and *TGF-β1* of the 3D HRPE spheroids were substantially higher than those of the 2D HRPE cells. While in contrast, mRNA expressions of *AGT* or *MYC* of the 3D HRPE spheroids were significantly increased under normoxia conditions or decreased, respectively, as compared to the 2D HRPE cells. These results rationally indicated that *STAT3*, *IL6*, *FOS* and *TGF-β1* were indeed pivotally involved in the generation processes of the 3D spheroids of HRPE cells, and these mechanisms may in turn affect the glycolysis and/or mitochondrial functions as above. Taking the above collective data into account along with the fact that each HIF1α and HIF2α regulates different functions of mitochondria, that is, respiration and oxidative stress, respectively [37], we conclude that TGF-β2, hypoxia, and culture conditions (2D or 3D) stimulate the EMT of the HRPE cells in different manners, presumably by mitochondria related cellular metabolisms.

## 3. Discussion

Epithelial mesenchymal transition (EMT) is a pivotal biological process, in which epithelial cells transdifferentiate into mesenchymal cells, and occurs during physiological conditions such as during normal embryonic development and wound healing as well as in several pathologic conditions such as fibrosis, cancer progression, and others [38,39]. There are three distinct subtypes of EMT: type 1 occurs during tissue and embryo development, type 2 is involved in wound healing and organ fibrosis, and type 3 is associated with cancer progression and metastasis [38,39]. Among these, it is well known that the type 2 EMT of RPE is mainly involved several ocular diseases including PVR, ARMD and others, and those underlying molecular mechanisms have been extensively investigated so far. However, in contrast, only a few studies are available concerning the effects of hypoxia toward EMT of the RPE cells and metabolic plasticity during these EMT processes. In the current study, to study this insufficiently identified issue, we examined the effects of hypoxia on the TGF-β2-induced EMT of the 2D and 3D cultured HRPE cells, and following results were obtained: (1) TGF-β2 and hypoxia significantly and synergistically increased the barrier function of the 2D HRPE monolayers, as evidence by TEER measurements, downsizing, the stiffening of the 3D HRPE spheroids and the mRNA expression of most of the ECM proteins; (2) a real-time metabolic analysis indicated that TGF-β2 caused the deterioration in both maximal mitochondrial oxidative phosphorylation capacity and glycolytic reserve in the 2D HRPE cells or caused an increased proton leakage of the 3D HRPE spheroids, respectively; (3) the mRNA expression of HIF1α, but not HIF2α, were synergistically up-regulated by TGF-β2 and hypoxia in the 2D HRPE cells, but this was not observed in the case of the 3D HRPE spheroids; and (4) the gene expressions of some of the recently determined possible up-stream regulators required for the generation of the 3D spheroids; *STAT3*, *IL6*, *FOS* and *TGF-β1*, and the mitochondria regulatory gene; *PGC1a* were substantially higher in the 3D HRPE spheroids compared to the 2D HRPE cells.

When oxygen levels are decreased (hypoxia), it is well known that three major physiological processes are evoked, namely, (1) blood is shunt within the lung to obtain as much oxygen as possible, (2) neurotransmitters are released to increase respiration, and (3) the production of erythropoietin (EPO) occurs, resulting in an increase in the hemoglobin concentration within the blood [40,41]. On the other hand, hypoxia is also pivotally involved in the etiology of many diseases [42], and within their underlying mechanisms, HIF1, a nuclear factor bound to a cis-acting hypoxia response element (HRE) was identified [43]. HIF1 is a heterodimer of HIF1α and HIF1β, in which only HIF1α is detectable under hypoxia conditions, despite the fact that the HIF1β subunit is stable [44]. HIF2α and HIF3α were also subsequently found to be involved in similar hypoxia-related regulation mechanisms [45,46]. It has also been suggested that such HIF-induced mechanisms may well be involved in the pathogenesis of EMT in RPE related retinal diseases. For example, in a laser CNV mouse model, the knockdown of HIF1α within RPE cells inhibited the overexpression of VEGF and intercellular adhesion molecule 1 (ICAM-1), thereby substantially reducing vascular leakage and the CNV area [47]. These findings reported herein also demonstrate that significantly higher amounts of HIF1α are produced by ARPE-19 cells under hypoxia conditions compared to under normoxia conditions. Therefore, these results indicate that HIF1α derived from RPE cells could be a possible stimulator of CNV progression by inducing the transcription of VEGF and ICAM-1. Indeed, such an HIF1α linked stimulation of the angiogenesis was also recognized as a pivotal mechanism that is associated with the molecular pathogenesis of diabetic retinopathy [48,49,50]. Furthermore, HIF1α also promotes the TGF-β2-induced EMT of human lens epithelial cells [51] as well as that for ARPE-19 cells [52]. As another important biological aspect of HIFs, it was quite interestingly revealed that HIFs regulate both mitochondrial respiration and mitochondrial oxidative stress, and conversely, mitochondrial metabolism, respiration and oxidative stress also could regulate the activation of HIFs [37]. In fact, Shu et al. reported that the inhibition of PGC1α, a master regulator of mitochondrial function, induced the disruption of the mitochondrial functions and the stimulation of an EMT response within human RPE cells [2]. In their subsequent study, they also demonstrated that TGF-β2 stimulated EMT in RPE, caused the significant down-regulation of PGC1α, and mitochondrial dysfunctions as well as a metabolic shift towards reduced OXPHOS and increased glycolysis [16]. In the current study, although we also found that TGF-β2 and hypoxia synergistically and differently induced EMT of the HRPE cells considering the fluctuation in the mRNA expression of HIF1α and the mitochondrial metabolism described above, those fluctuations were also different between 2D and 3D cultures, and such diversity between them may be caused by possible up-stream regulators requiring the generation of the 3D spheroids, as was recently determined using 3T3-L1 cells [36].

However, the underlying mechanisms responsible causing the synergistic effect on mitochondrial metabolisms by hypoxia and TGF-β2 during EMT of the HRPE cells remain insufficiently explained. Therefore, to better understand these unidentified issues, further investigations, including an RNA-Seq experiment will be required as our next project.

## 4. Methods

### 4.1. Two-Dimensional and Three-Dimensional Cultures of Human Retinal Pigment Epithelium (HRPE) Cells

All experiments using human derived cells were conducted in compliance with the tenets of the Declaration of Helsinki after approval by the internal review board of Sapporo Medical University. A commercially available human retinal pigment epithelium cell line ARPE-19 purchased from the American Type Culture Collection (ATCC, #CRL-2302™, Manassas, VA, USA) was cultured in 150 mm 2D culture dishes until they reached 90% confluence at 37 °C in grown medium A composed of HG-DMEM containing 10% FBS, 1% L-glutamine, 1% antibiotic-antimycotic and were maintained by changing the medium every other day under standard normoxia conditions (37 °C, 5% CO_2_) or hypoxia conditions (37 °C, 5% CO_2_, 1% O_2_). For measurement of cell viability, a CCK-8 kit (DOJINDO, Osaka, Japan) was used. 2D HRPE cells were seeded in a 96-well plate and incubated at 37 °C. When the cell density reached 80–90% of the well, the cells were washed with phosphate buffered saline (PBS) and treated with CCK-8 reagent for 1 h at 37 °C. Absorbance was measured at 450 nm using a microplate reader (EnSpire, Perkinelmer, Waltham, MA, USA). Experiments were taken in six parallel wells and repeated in dupicate. The TEER values for the HRPE cell monolayers were determined using a TEER plate (0.4 μm pore size and 12 mm diameter; Corning Transwell, Sigma-Aldrich, Burlington, MA, USA) and an electrical resistance system (KANTO CHEMICAL CO. INC., Tokyo, Japan) as described in a previous study [32,53]. Alternatively, HRPE cells prepared as above were further processed for a 3D spheroid preparation, as described below.

The 3D spheroids of HRPE were generated in a hanging droplet spheroid three-dimension (3D) culture system, basically as described in a previous our report in which human trabecular meshwork (HTM) cells [31]. Briefly, 90% confluence HRPE cells in 150 mm 2D culture dishes as above were washed with PBS, and the cells were detached by treatment with 0.25% Trypsin/EDTA. After centrifugation for 5 min at 300× *g*, the cell pellet was re-suspended in spheroid medium A composed of growth medium A supplemented with 0.25% methylcellulose (Methocel A4M) to facilitate the formation of a stable 3D spheroid morphology. Approximately 20,000 HRPE cells in the 28 μL of spheroid medium A were placed into each well of a hanging drop culture plate (#HDP1385, Sigma-Aldrich) (Day 0). At Day 1, 5 ng/mL TGFβ2 was added to the spheroid medium A to stimulate epithelial-mesenchymal transition (EMT). On each following day, a half of the spheroid medium was replaced by fresh medium and the cultures were maintained until Day 6.

### 4.2. Measurement of the Size and Solidity of 3D HRPE Spheroids

For evaluating physical properties, the mean size and stiffness, of the 3D HRPE spheroids were determined by measuring their largest cross-sectional area (CSA) using an inverted microscope (Nikon ECLIPSE TS2; Tokyo, Japan), a micro-squeezer (MicroSquisher, CellScale, Waterloo, ON, Canada) as reported in a previous study [29,31].

### 4.3. Immunocytochemistry of 2D and 3D Cultured HRPE Cells

The immunocytochemistry of the 2D and 3D cultured HRPE cells as above was evaluated using 1st antibodies; an anti-human rabbit antibody (1:200 dilutions) against COL1, COL4, COL6 (ROCKLAND antibodies & assays, Limerick, PA, USA) or FN (Santa Cruz Biotechnology, Inc., Dallas, TX, USA), a goat anti-rabbit IgG (488 nm, 1:1000 dilutions, Invitrogen, Waltham, MA, USA), and DAPI (1:1000 dilutions, DOJINDO, Osaka, Japan), phalloidin (594 nm, 1:1000 dilutions, Cayman Chemical, Ann Arbor, MI, USA) and DAPI (1:1000 dilutions, DOJINDO, Osaka, Japan), and confocal immunofluorescent images were obtained, as described in a recent report [30,34].

### 4.4. Measurement of Real-Time Cellular Metabolic Functions

The rates of oxygen consumption (OCR) and extracellular acidification (ECAR) of 2D and 3D cultured HRPE cells in the absence and presence of 5 ng/mL TGF-β2 under normoxia conditions were measured using a Seahorse XFe96 Bioanalyzer (Agilent Technologies) according to the manufacturer’s instructions. Briefly, approximately 20 × 10^3^ of 2D cultured cells were each placed in a well of a XFe96 Cell Culture Microplate (Agilent Technologies, #103794-100). Following centrifugation of the plate at 1600× *g* for 10 min, the culture medium was replaced with 180 μL of assay buffer (Seahorse XF DMEM assay medium (pH 7.4, Agilent Technologies, #103575-100), supplemented with 5.5 mM glucose, 2.0 mM glutamine, and 1.0 mM sodium pyruvate). For 3D cultured cells, spheroids were washed twice with PBS and three to five spheroids were placed onto a well of a XFe96 Spheroid Microplate (Agilent Technologies, #102978-100) containing 180 μL of assay buffer. The assay plates were incubated in a CO_2_-free incubator at 37 °C for 1 hour prior to the measurements. OCR and ECAR were measured using the Seahorse XFe96 Bioanalyzer under 3 min mix and 3 min measure protocols at baseline and following the injection of oligomycin (final concentration: 2.0 μM), carbonyl cyanide p-trifluoromethoxyphenylhydrazone (FCCP, final concentration: 5.0 μM), a mixture of rotenone/antimycin A (final concentration: 1.0 μM), and 2-deoxyglucose (2-DG, final concentration: 10 mM). Since the effects of drug injection are different between 2D and 3D conditions, 3 cycles of each measurement were employed for the 2D cells, and 8 cycles for a measurement with oligomycin and 4 cycles for other measurements were employed for the 3D cells. The OCR and ECAR values were normalized to the amount of protein per well in the 2D cells and to the number of spheroids in the 3D cells.

Baseline OCR was determined by subtracting the OCR with rotenone/antimycin A from the OCR at baseline. ATP-linked Respiration was determined by the difference in OCR after the addition of oligomycin. Proton Leakage was determined by subtracting OCR with rotenone/antimycin A from the OCR after the addition of oligomycin. Respiratory capacity was determined by subtracting OCR with rotenone/antimycin A from OCR after the addition of FCCP. Baseline ECAR was determined by subtracting ECAR with 2-DG from ECAR at the baseline. Glycolytic capacity was determined by subtracting ECAR with 2-DG from ECAR with oligomycin. Glycolytic reserve was determined by the difference in ECAR after the addition of oligomycin.

### 4.5. Other Analytical Methods

Total RNA was extracted from the 2D or 3D cultured HRPE cells as above and reverse transcription and real-time PCR were carried out as previously reported [30,34] using specific primers and probes (Appendix A).

All statistical analyses were performed using Graph Pad Prism 8 (GraphPad Software, San Diego, CA) as described in a recent report [30,34].

## Figures and Tables

**Figure 1 ijms-23-05473-f001:**
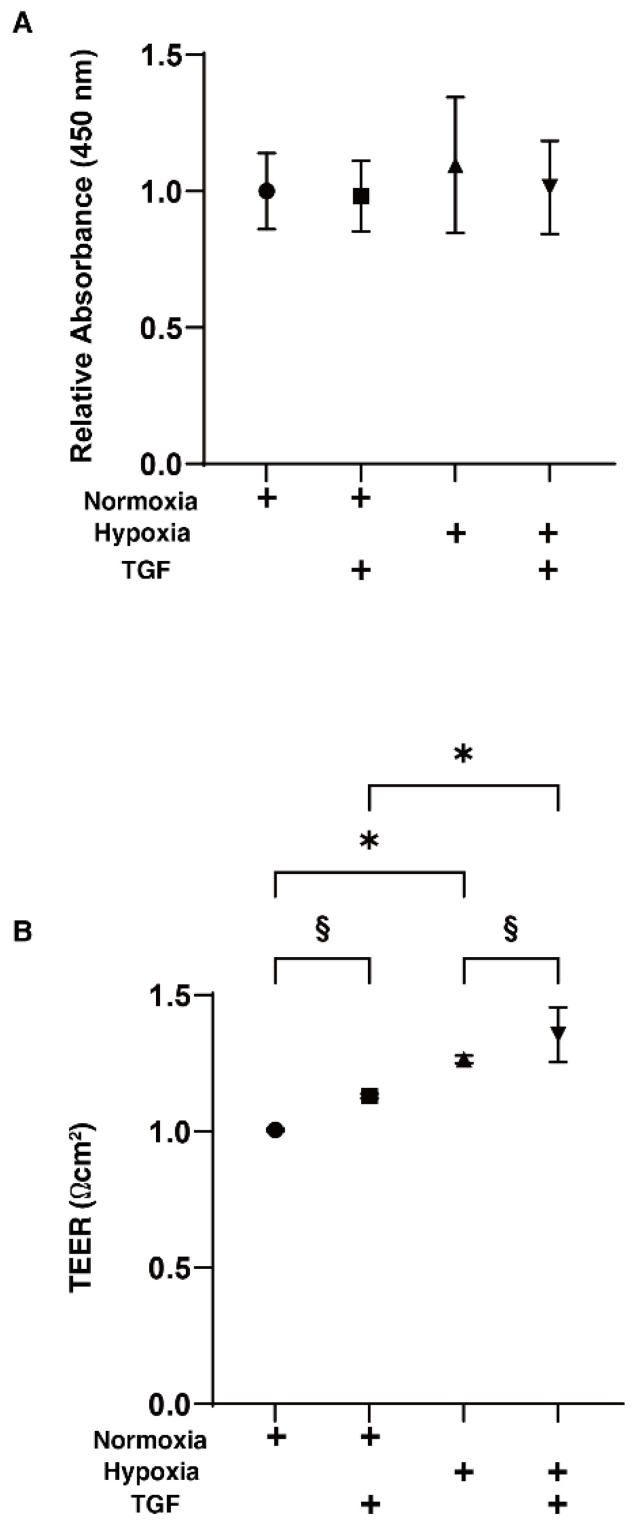
Effects of TGF-β2 on cell viability and transendothelial electrical resistance (TEER) measurements of 2D HRPE monolayers under normoxia or hypoxia conditions. The 2D HRPE cell monolayers prepared under normoxia or hypoxia conditions were treated without (control) or with a 5 ng/mL solution of TGF-β2 (TGF). The 2D cultures of HRPE monolayers at Day 6 were subjected to (**A**) cell viability by a CCK-8 kit and (**B**) barrier function analyses by electric resistance (Ω cm^2^) measurements using TEER, and each relative ratio against the normoxia conditions without TGF-β2 was plotted. All experiments were performed in triplicate using fresh preparations. “+” is reagents addition. Data are presented as the arithmetic mean ± the standard error of the mean (SEM). * *p* < 0.05 normoxia vs. hypoxia, § *p* < 0.05 control vs. TGF (ANOVA followed by a Tukey’s multiple comparison test).

**Figure 2 ijms-23-05473-f002:**
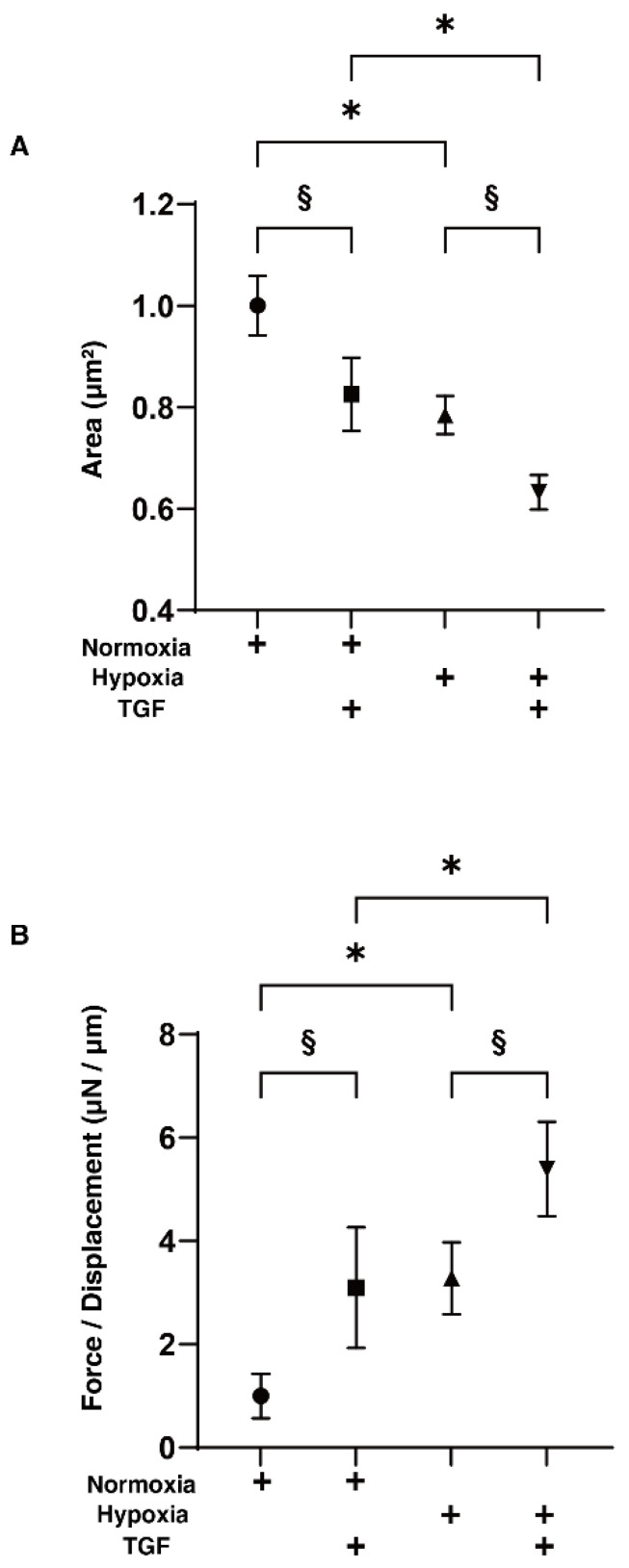
Effects of TGF-β2 on the physical properties, sizes and stiffness of 3D HRPE spheroids under normoxia or hypoxia conditions. Three-dimensional HRPE spheroids at Day 6 prepared under normoxia or hypoxia conditions were treated without (control) or with a 5 ng/mL solution of TGF-β2 (TGF). The mean sizes of 3D HRPE spheroids were measured and each relative ratio against the normoxia conditions without TGF-β2 are plotted in panel (**A**). The physical solidity of the 3D HRPE spheroids was analyzed by a micro-squeezer and the force required to produce a 50% deformity of a single spheroid during a period of 20 s (μN/μm) was plotted in panel (**B**). All experiments were performed in triplicate using fresh preparations consisting of 16 spheroids each. “+” is reagents addition. Data are presented as the arithmetic mean ± standard error of the mean (SEM). * *p* < 0.05 control vs. TGF, § *p* < 0.05 normoxia vs. hypoxia (ANOVA followed by a Tukey’s multiple comparison test).

**Figure 3 ijms-23-05473-f003:**
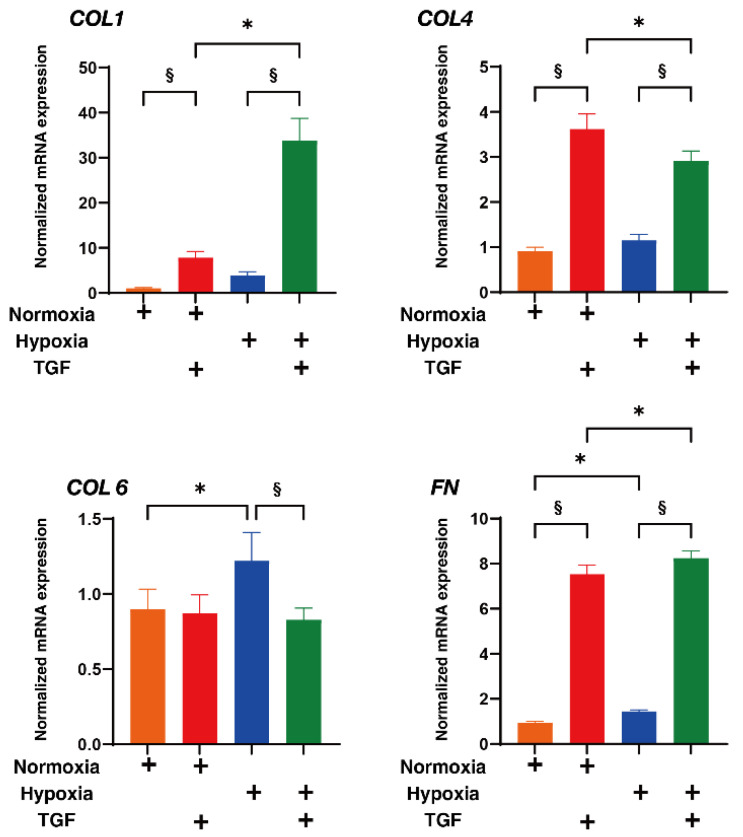
Effects of TGF-β2 on the mRNA expression of ECMs of 2D cultured HRPE cells under normoxia or hypoxia conditions. 2D cultured HRPE cells at Day 6 prepared under normoxia or hypoxia conditions were treated without (control) or with a 5 ng/mL solution of TGF-β2 (TGF), and each sample was subjected to qPCR analysis and the expression of mRNA in ECMs, *COL1*, *COL4*, *COL6*, and *FN* were estimated. All experiments were performed in duplicate using fresh preparations (n = 5). “+” is reagents addition. Data are presented as the arithmetic mean ± the standard error of the mean (SEM). * *p* < 0.05 control vs. TGF, § *p* < 0.05 normoxia vs. hypoxia (ANOVA followed by a Tukey’s multiple comparison test).

**Figure 4 ijms-23-05473-f004:**
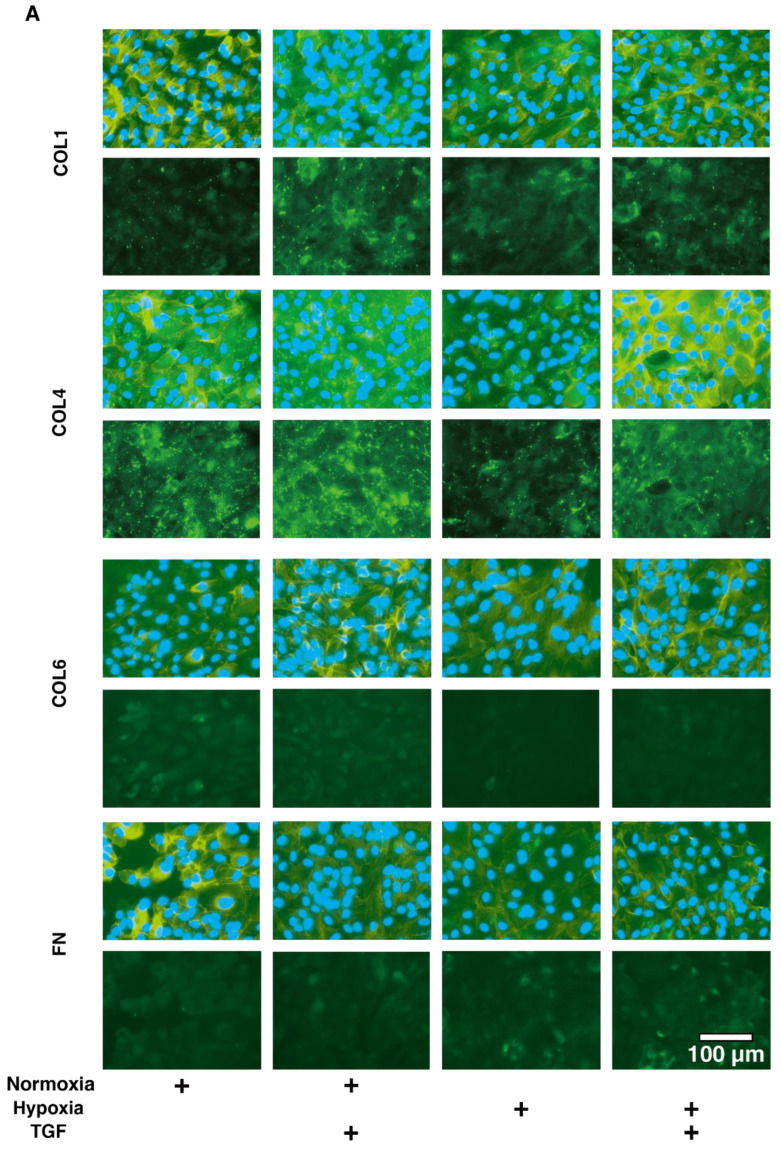
Immunolabeling of ECMs of the 2D cultured HRPE cells under normoxia or hypoxia conditions. 2D cultured HRPE cells at Day 6 prepared under normoxia or hypoxia conditions were treated without (control) or with a 5 ng/mL solution of TGF-β2 (TGF), and each sample was subjected to immunostaining for *COL1*, *COL4*, *COL6*, and *FN*. All experiments were performed in duplicate using fresh preparations (n = 5). Representative images (upper; merge with DAPI, lower; anti-ECM, scale bar; 100 μm) are shown in panel (**A**) and relative staining intensities against the normoxia conditions without TGF-β2 are plotted in panel (**B**). “+” is reagents addition. Data are presented as the arithmetic mean ± standard error of the mean (SEM). * *p* < 0.05 control vs. TGF, § *p* < 0.05 normoxia vs. hypoxia (ANOVA followed by a Tukey’s multiple comparison test).

**Figure 5 ijms-23-05473-f005:**
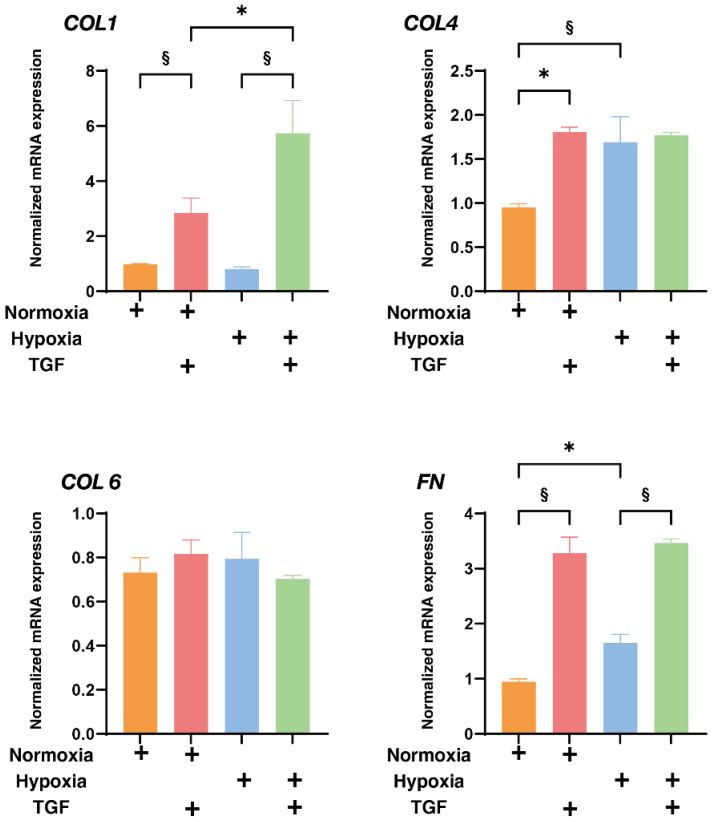
Effects of TGF-β2 on mRNA expression of ECMs of 3D HRPE spheroids under normoxia or hypoxia conditions. Three-dimensional HRPE spheroids at Day 6 prepared under normoxia or hypoxia conditions were treated without (control) and with a 5 ng/mL solution of TGF-β2 (TGFβ), and each sample was subjected to qPCR analysis and the expression of mRNA in ECMs, *COL1*, *COL4*, *COL6*, and *FN* were estimated. All experiments were performed in duplicate using fresh preparations (n = 10 spheroids each). “+” is reagents addition. Data are presented as the arithmetic mean ± the standard error of the mean (SEM). * *p* < 0.05 control vs. TGF, § *p* < 0.05 normoxia vs. hypoxia (ANOVA followed by a Tukey’s multiple comparison test).

**Figure 6 ijms-23-05473-f006:**
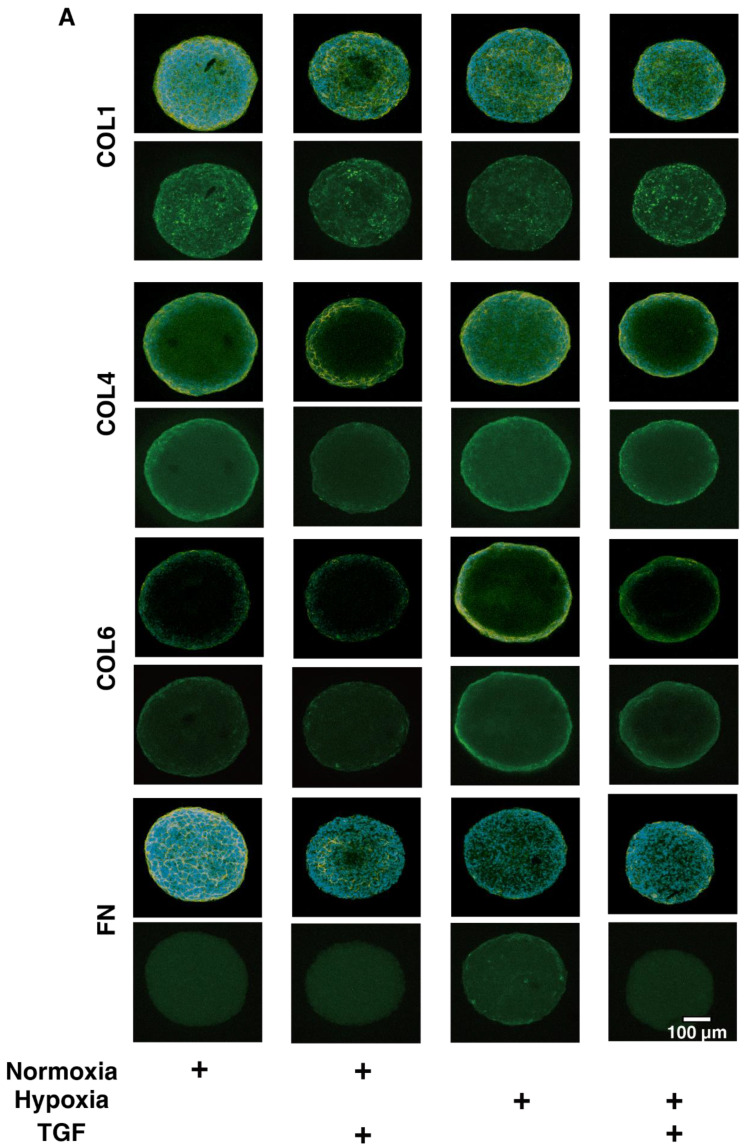
Immunolabeling of ECMs of the 3D HRPE spheroids under normoxia or hypoxia conditions. Three-dimensional HRPE spheroids at Day 6 prepared under normoxia or hypoxia conditions were treated without (control) and with a 5 ng/mL solution of TGF-β2 (TGF), and each sample was then subjected to immunostaining for *COL1*, *COL4*, *COL6*, and *FN*. All experiments were performed in duplicate using fresh preparations (n = 5 spheroids each). Representative images (upper; merged with DAPI, lower; anti-ECM, scale bar; 100 μm) are shown in panel A and relative staining intensities against the normoxia conditions without TGF-β2 were plotted in panel B. “+” is reagents addition. Data are presented as the arithmetic mean ± standard error of the mean (SEM). * *p* < 0.05 control vs. TGF, § *p* < 0.05 normoxia vs. hypoxia (ANOVA followed by a Tukey’s multiple comparison test).

**Figure 7 ijms-23-05473-f007:**
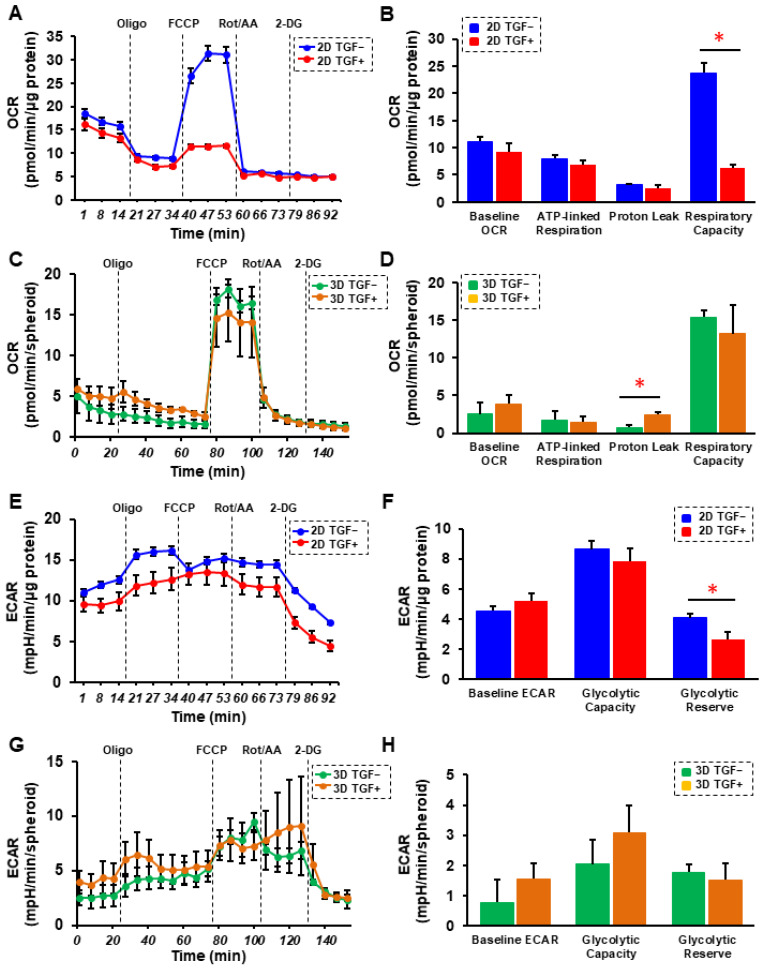
Effects of TGF-β2 on the cellular metabolic phenotype of the 2D and 3D cultured HRPE cells. Two-dimensional and 3D cultured HRPE cells at Day 6 prepared under normoxia conditions were treated without (control) and with a 5 ng/mL solution of TGF-β2 (TGF), and each sample was subjected to a real-time metabolic function analysis using a Seahorse XFe96 Bioanalyzer. (**A**) Measurements of oxygen consumption rate (OCR) in 2D HRPE cells without and with TGF treatment. (**B**) Baseline OCR, ATP-linked respiration, proton leak, and respiratory capacity in 2D HRPE cells. (**C**) Measurements of oxygen consumption rate (OCR) in 3D HRPE spheroids without and with TGF. (**D**) Baseline OCR, ATP-linked respiration, proton leak, and respiratory capacity in 3D HRPE spheroids. (**E**) Measurements of extracellular acidification rate (ECAR) in 2D HRPE cells without and with TGF. (**F**) Baseline ECAR, glycolytic capacity, glycolytic reserve in 2D HRPE cells. (**G**) Measurements of extracellular acidification rate (ECAR) in 3D HRPE spheroids without or with TGF. (**H**) Baseline ECAR, glycolytic capacity, glycolytic Reserve in 3D HRPE spheroids. Oligo = oligomycin, Rot/AA = rotenone/antimycin A, 2-DG = 2-deoxyglucose. Fresh preparations were used in all experiments (2D; n = 5, 3D; n = 3). Data are presented as the mean ± the standard error of the mean (SEM). * *p* < 0.05 control vs. TGF (Student’s *t*-test).

**Figure 8 ijms-23-05473-f008:**
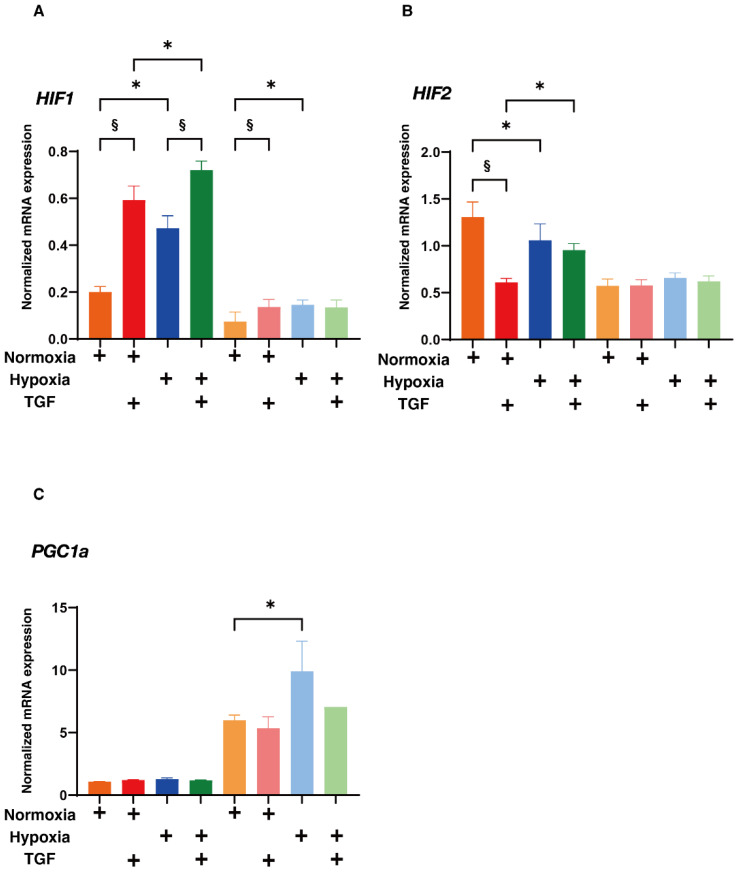
Effects of TGF-β2 on the mRNA expression of hypoxia-related factors; (**A**,**B**) HIF1α and 2α and mitochondria regulatory gene; (**C**) *PGC1a* of 2D and 3D HRPE cells under normoxia or hypoxia conditions. Two-dimensional and three-dimensional cultured HRPE cells at Day 6 prepared under normoxia and hypoxia conditions were treated without (control) or with a 5 ng/mL solution of TGF-β2 (TGFβ), and each sample was subjected to qPCR analysis and the expression of mRNA in hypoxia-related factors; *HIF1α* and *2α* and mitochondria regulatory gene; *PGC1a* were estimated. All experiments were performed in duplicate using fresh preparations (2D; n = 5, 3D; n = 10 spheroids each). “+” is reagents addition. Data are presented as the arithmetic mean ± the standard error of the mean (SEM). * *p* < 0.05 control vs. TGF, § *p* < 0.05 normoxia vs. hypoxia (ANOVA followed by a Tukey’s multiple comparison test).

**Figure 9 ijms-23-05473-f009:**
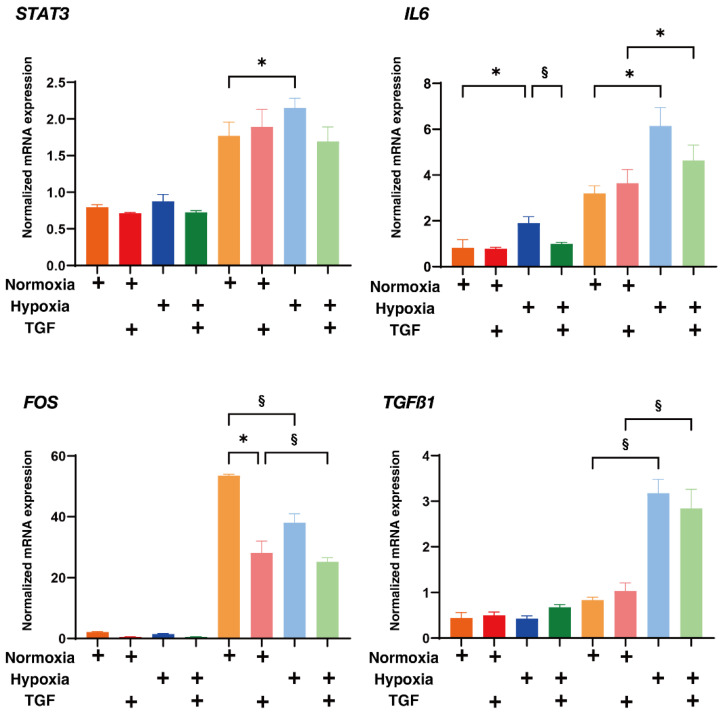
Effects of TGF-β2 on mRNA expression of the possible up-stream regulators of 3D spheroids; *STAT3*, *IL6*, *FOS*, *TGFb1*, *AGT* and *MYC* of 2D and 3D HRPE cells under normoxia or hypoxia conditions. Two-dimensional and three-dimensional cultured HRPE cells at Day 6 prepared under normoxia or hypoxia conditions were treated without (control) or with a 5 ng/mL solution of TGF-β2 (TGFβ), and each sample was subjected to qPCR analysis and the expression of mRNA in the possible up-stream regulators requiring for the generation of the 3D spheroids [36]; *STAT3*, *IL6*, *FOS*, *TGFb1*, *AGT* and *MYC*, were estimated. All experiments were performed in duplicate using fresh preparations (2D; n = 5, 3D; n = 10 spheroids each). “+” is reagents addition. Data are presented as the arithmetic mean ± the standard error of the mean (SEM). * *p* < 0.05 control vs. TGF, § *p* < 0.05 normoxia vs. hypoxia (ANOVA followed by a Tukey’s multiple comparison test).

## Data Availability

The data that support the findings of this study are available from the corresponding author upon reasonable request.

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
