# Peer review of "Hypoxia Differently Affects TGF-β2-Induced Epithelial Mesenchymal Transitions in the 2D and 3D Culture of the Human Retinal Pigment Epithelium Cells"

_ijms, 2022, doi:10.3390/ijms23105473_

Round 1
Reviewer 1 Report
This study tested the effect of TGF-beta and hypoxia on the epithelial-mesenchymal transition (EMT) of human RPE cells (ARPE19 cell line) in 2D and 3D culture. The authors found significant differences in the response of 2D and 3D cultures to hypoxia and TGF-beta. TGF-beta-induced EMT-associated mitochondrial metabolism of both 2D and 3D cultured cells, but the effects were different in 2D versus 3D cultures. TGF-beta increased barrier properties in both culture types. TGF-beta and hypoxia upregulated different genes in 2D and 3D cultures.
General comments:
This is an interesting study about the differential response of 2D versus 3D cultures to TGF-beta and hypoxia. However, the presentation of the figures should be improved.
Phalloidin (red Actin stain in confocal images): should be explained in figure legends (Figure 4A, 6A). The significance of the differential Actin staining between experimental group should also be mentioned in the discussion.
Specific comments:
Results:
- 4, bottom paragraph, 3rd line: There should be an “and” between “qPCR analysis” and “by”.
- 8 bottom line – p.9, 1st line: “2D-spheroids” – should be “2D culture”
Discussion:
- 12, 2nd line of 2nd paragraph: “three major physiological are evoked” – should be “three major physiological processes …”
- 12, bottom paragraph: “the underlying mechanisms responsible … remain to be insufficient” – should be “… remain insufficiently explained”
Figures:
The black and white graphs in Figs. 3, 4B, 5, 6B, 8 and 9 would look better in color. It would help to give each experimental group a specific color code.
Figure 4A, Figure 6A: confocal image panels are too small (too low resolution).
Figure 8: there should be A, B, C. The panel for HIF2 is too close to the one for HIF1. The HIF2 heading should move over the Y-axis as it is on the other panels.
Author Response
Dear Editor,
Thank you very much for the constructive comments concerning our manuscript, " Hypoxia differently affects TGF-b2 induced epithelial mesenchymal transitions in the 2D and 3D culture of the human retinal pigment epithelium cells”. We examined the Reviewer's comments carefully and prepared a revised version of our paper that takes these comments into account. The changes are listed below.
Reviewer 1
This study tested the effect of TGF-beta and hypoxia on the epithelial-mesenchymal transition (EMT) of human RPE cells (ARPE19 cell line) in 2D and 3D culture. The authors found significant differences in the response of 2D and 3D cultures to hypoxia and TGF-beta. TGF-beta-induced EMT-associated mitochondrial metabolism of both 2D and 3D cultured cells, but the effects were different in 2D versus 3D cultures. TGF-beta increased barrier properties in both culture types. TGF-beta and hypoxia upregulated different genes in 2D and 3D cultures.
General comments:
This is an interesting study about the differential response of 2D versus 3D cultures to TGF-beta and hypoxia. However, the presentation of the figures should be improved.
Phalloidin (red Actin stain in confocal images): should be explained in figure legends (Figure 4A, 6A). The significance of the differential Actin staining between experimental group should also be mentioned in the discussion.
Answer; In terms of the presentation of the figures, another reviewer also commented on the use of statistical symbols, in addition to several comments as below. Therefore, all figures were revised as follows according these comments and also just one p value level i.e. p<0.05 and just one symbol for relevant dicotomic comparison: e.g * p<0.05 control vs hypoxia, § p<0.05 control vs TGFbeta was used without comparison between 2D and 3D cells as suggested by the reviewer 2. In terms of phalloidin staining within Figs. 4 and 6, those intensities of the merged images were significant influenced and linked with the staining intensities of the target ECM proteins. Thus, this may mislead the significance of the differential actin staining between experimental groups. Therefore, to avoid such misleading, the phalloidin staining was removed.
Specific comments:
Results:
- 4, bottom paragraph, 3rd line: There should be an “and” between “qPCR analysis” and “by”.
Answer; As pointed out, those were changed to “An analysis by qPCR”.
- 8 bottom line – p.9, 1st line: “2D-spheroids” – should be “2D culture”
Answer; As pointed out, those were changed to “in the 2D HRPE cultures rather than the 3D HRPE spheroid cultures”.
Discussion:
- 12, 2nd line of 2nd paragraph: “three major physiological are evoked” – should be “three major physiological processes …”
Answer; As pointed out, those were changed to “three major physiological processes”.
- 12, bottom paragraph: “the underlying mechanisms responsible … remain to be insufficient” – should be “… remain insufficiently explained”
Answer; As pointed out, those were changed to “remain insufficiently explained”
Figures:
- The black and white graphs in Figs. 3, 4B, 5, 6B, 8 and 9 would look better in color. It would help to give each experimental group a specific color code.
Answer; As pointed out, those graphs were changed to color coded bars rather than black and white bars.
- Figure 4A, Figure 6A: confocal image panels are too small (too low resolution).
Answer; As pointed out, to make these confocal images more visible, these figures were revised.
- Figure 8: there should be A, B, C. The panel for HIF2 is too close to the one for HIF1. The HIF2 heading should move over the Y-axis as it is on the other panels.
Answer; As pointed out, we added labels A, B and C and heading of HIF2 is moved as similarly to others.

Reviewer 2 Report
The manuscript is well written. The authors explored effects of TGFbeta2 on immortalized RPE monolayer and spheroids. Abstract should be revised, instead of methods, authors are invited to describe the background of research in order to highlight the experimental hypothesis. Authors must justify in the introduction the use of spheroids, along with 2D monolayer, and any putative advantage in terms of translational research. As regards as results, authors must justify the increase of TEER after hypoxia and tgfbeta2 treatment, since it is expected that hypoxia would damage the 2D ARPE monolayer as simple in-vitro model of oBRB. Immunocytochemistry images are at very low resolution and need to be totally revised. As regards as statistical analysis and use of statistical symbols, authors are invited to totally revise the representation of statistics. Authors used too many symbols and figures, such as qPCR figure, are too messy. Therefore, authors are invited to use just one p value level i.e. p<0.05 and just one symbol for relevant dicotomic comparison: e.g * p<0.05 control vs hypoxia, § p<0.05 control vs TGFbeta and so on. Comparison between 2D and 3D cells in figure 8 and 9 are not relevant, and too many statistical symbol are currently used. Please revise all the figures. As regards as the TGFbeta signaling pathway-hypoxia-fibrosis authors should discuss their results on the basis of current literature regarding diabetic retinopathy and putative implications at oBRB (PMID: 32848728, PMID: 33334029, PMID: 30222965)
Author Response
Dear Editor,
Thank you very much for the constructive comments concerning our manuscript, " Hypoxia differently affects TGF-b2 induced epithelial mesenchymal transitions in the 2D and 3D culture of the human retinal pigment epithelium cells”. We examined the Reviewer's comments carefully and prepared a revised version of our paper that takes these comments into account. The changes are listed below.
Reviewer 1
This study tested the effect of TGF-beta and hypoxia on the epithelial-mesenchymal transition (EMT) of human RPE cells (ARPE19 cell line) in 2D and 3D culture. The authors found significant differences in the response of 2D and 3D cultures to hypoxia and TGF-beta. TGF-beta-induced EMT-associated mitochondrial metabolism of both 2D and 3D cultured cells, but the effects were different in 2D versus 3D cultures. TGF-beta increased barrier properties in both culture types. TGF-beta and hypoxia upregulated different genes in 2D and 3D cultures.
General comments:
This is an interesting study about the differential response of 2D versus 3D cultures to TGF-beta and hypoxia. However, the presentation of the figures should be improved.
Phalloidin (red Actin stain in confocal images): should be explained in figure legends (Figure 4A, 6A). The significance of the differential Actin staining between experimental group should also be mentioned in the discussion.
Answer; In terms of the presentation of the figures, another reviewer also commented on the use of statistical symbols, in addition to several comments as below. Therefore, all figures were revised as follows according these comments and also just one p value level i.e. p<0.05 and just one symbol for relevant dicotomic comparison: e.g * p<0.05 control vs hypoxia, § p<0.05 control vs TGFbeta was used without comparison between 2D and 3D cells as suggested by the reviewer 2. In terms of phalloidin staining within Figs. 4 and 6, those intensities of the merged images were significant influenced and linked with the staining intensities of the target ECM proteins. Thus, this may mislead the significance of the differential actin staining between experimental groups. Therefore, to avoid such misleading, the phalloidin staining was removed.
Specific comments:
Results:
- 4, bottom paragraph, 3rd line: There should be an “and” between “qPCR analysis” and “by”.
Answer; As pointed out, those were changed to “An analysis by qPCR”.
- 8 bottom line – p.9, 1st line: “2D-spheroids” – should be “2D culture”
Answer; As pointed out, those were changed to “in the 2D HRPE cultures rather than the 3D HRPE spheroid cultures”.
Discussion:
- 12, 2nd line of 2nd paragraph: “three major physiological are evoked” – should be “three major physiological processes …”
Answer; As pointed out, those were changed to “three major physiological processes”.
- 12, bottom paragraph: “the underlying mechanisms responsible … remain to be insufficient” – should be “… remain insufficiently explained”
Answer; As pointed out, those were changed to “remain insufficiently explained”
Figures:
- The black and white graphs in Figs. 3, 4B, 5, 6B, 8 and 9 would look better in color. It would help to give each experimental group a specific color code.
Answer; As pointed out, those graphs were changed to color coded bars rather than black and white bars.
- Figure 4A, Figure 6A: confocal image panels are too small (too low resolution).
Answer; As pointed out, to make these confocal images more visible, these figures were revised.
- Figure 8: there should be A, B, C. The panel for HIF2 is too close to the one for HIF1. The HIF2 heading should move over the Y-axis as it is on the other panels.
Answer; As pointed out, we added labels A, B and C and heading of HIF2 is moved as similarly to others.
Reviewer 2
- The manuscript is well written. The authors explored effects of TGFbeta2 on immortalized RPE monolayer and spheroids. Abstract should be revised, instead of methods, authors are invited to describe the background of research in order to highlight the experimental hypothesis.
Answer; Thank you so much for this information. As suggested, instead of describing methods, the background of research in order to highlight the experimental hypothesis is included within the Abstract and therefore the abstract was revised to a more compact size; “The hypoxia associated with the transforming growth factor-β2 (TGF-β2)-induced epithelial mesenchymal transition (EMT) of human retinal pigment epithelium (HRPE) cells is well recognized as the essential underlying mechanism responsible for the development of proliferative retinal diseases. In vitro, three-dimension (3D) models associated with spontaneous O2 gradients can be used to recapitulate the pathological levels of hypoxia. to study the effect of hypoxia on the TGF-β2-induced EMT of HRPE cells in detail, we used two-dimension (2D) and 3D cultured HRPE cells. TGF-β2 and hypoxia significantly and synergistically increased the barrier function of the 2D HRPE monolayers, as evidenced by TEER measurements, the down-sizing and stiffening of the 3D HRPE spheroids and the mRNA expression of most of the ECM proteins. A real-time metabolic analysis indicated that TGF-β2 caused a decrease in the maximal capacity of mitochondrial oxidative phosphorylation in the 2D HRPE cells, whereas, in the case of 3D HRPE spheroids, TGF-β2 increased proton leakage. The findings reported herein indicate that the TGF-b2 induced EMT of both the 2D and 3D cultured HRPE cells were greatly modified by hypoxia, but during these EMT processes, the metabolic plasticity was different between 2D and 3D HRPE cells, suggesting that the mechanisms responsible for the EMT of the HRPE cells may be variable during their spatial spreading.”
- Authors must justify in the introduction the use of spheroids, along with 2D monolayer, and any putative advantage in terms of translational research.
Answer; Thank you so much for this information. As suggested, this justification is now included within 3rd paragraph of Introduction; “During the pathological progression of EMR in RPE cells, the resulting RPE fibrosis associated with neovascular vessels spatially spread toward both the neural retina and choroid. For example, in the case of the neovascular ARMD (neovARMD), neovessels sprout in either the choroidal or the subretinal directions to form choroidal neovascular membranes (CNV) or intraretinal angiomatous proliferations within the macular region [17]. The results of histological studies indicate that several cells including RPE, vascular endothelial cells, macrophages, pericytes, fibroblastic cells and myofibroblasts in addition to ECM proteins are present within the CNV membranes obtained from patients with neovARMD [18-20]. Furthermore, during the progression of this disease, it was histologically observed that degenerating RPE cells were reversibly transformed into mesenchymal cells via EMT to adapt such a harsh microenvironment [21-24]. Therefore, to understand the molecular mechanisms responsible for the spatially growing EMT changes in RPE cells, it becomes necessary to study the effects of both TGF-b2 and hypoxia, as possible major pathogenic inducers using a reliable 3D cell culture system that replicates such a spatial environment. In fact, in addition to the conventional 2D cell cultures, such an in vitro 3D spheroid model has been identified as a powerful tool for studying the molecular mechanisms of EMT as well as hypoxia within cancerous cells [25-28].
- As regards as results, authors must justify the increase of TEER after hypoxia and tgfbeta2 treatment, since it is expected that hypoxia would damage the 2D ARPE monolayer as simple in-vitro model of oBRB.
Answer; Thank you so much for this information. As suggested, this justification is now included within the 1st paragraph of Results section of the paper; “To study the effects of hypoxia on the TGF-b2 induced epithelial-mesenchymal transition (EMT) of the retinal pigment epithelium (RPE), 2D and 3D cultured HRPE cells were prepared in the absence and presence of a 5 ng/ml solution of TGF-b2 under normoxia and under hypoxia conditions. Prior to the current analytical experiments, a cell viability assay confirmed that these experimental conditions were not toxic (Fig. 1A). To evaluate the function of the putative outer Blood Retinal Barrier (oBRB) that is comprised of RPE cells, we used the 2D HRPE monolayers as a simple in vitro model of oBRB. Within the 2D HRPE monolayers, the TGF-b2 treatment and hypoxia resulted in a significant increase in the TEER values of those untreated under normoxia conditions, and both types of stimulations synergistically induced further enhancement effects (Fig. 1B). Similar to this, a mono-treatment with TGF-b2 or hypoxia also caused a significant decrease and increase of the mean sizes and stiffness of the 3D HRPE spheroids, respectively, and such effects were also synergistically enhanced by the simultaneous treatments of both stimulants (Fig. 2).”.
- Immunocytochemistry images are at very low resolution and need to be totally revised.
Answer; As pointed out, to make these confocal images more visible, these figures were revised.
- As regards as statistical analysis and use of statistical symbols, authors are invited to totally revise the representation of statistics. Authors used too many symbols and figures, such as qPCR figure, are too messy. Therefore, authors are invited to use just one p value level i.e. p<0.05 and just one symbol for relevant dicotomic comparison: e.g * p<0.05 control vs hypoxia, § p<0.05 control vs TGFbeta and so on. Comparison between 2D and 3D cells in figure 8 and 9 are not relevant, and too many statistical symbol are currently used. Please revise all the figures.
Answer; As pointed out, to make these statistical symbols within these figures more simpler, the figures were revised to use just one p value level i.e. p<0.05 and just one symbol for relevant dicotomic comparison: * p<0.05 control vs TGFbeta, § p<0.05 normoxia vs hypoxia without comparison between 2D and 3D cells.
- As regards as the TGFbeta signaling pathway-hypoxia-fibrosis authors should discuss their results on the basis of current literature regarding diabetic retinopathy and putative implications at oBRB (PMID: 32848728, PMID: 33334029, PMID: 30222965)
Answer; Thank you so much for providing us with this important information related with HIF1alpha within the diabetic retinopathy. Therefore, this information is now included in the 2nd paragraph of the Discusion section; “When oxygen levels are decreased (hypoxia), it is well known that three major physiological processes are evoked, that is, 1) blood is shunt within the lung to get as much oxygen as possible, 2) neurotransmitters are released to increase respiration, and 3) production of erythropoietin (EPO) to increase the hemoglobin concentration within the blood occurs [37,38]. On the other hand, hypoxia is also pivotally involved in the etiology of many diseases [39], and within their underlying mechanisms, HIF1, a nuclear factor bound to a cis-acting hypoxia response element (HRE) was identified [40]. HIF1 is a heterodimer of HIF1α and HIF1β, in which only HIF1α is detectable under hypoxia conditions, despite the fact that the HIF1β subunit is stable [41]. HIF2α and HIF3α were also subsequently found to be involved in similar hypoxia related regulation mechanisms [42,43]. It has also been suggested that such HIF induced mechanisms may well be involved in the pathogenesis of EMT in RPE related retinal diseases. For example, in a laser CNV mouse model, the knockdown of HIF1α within RPE cells inhibited the overexpression of VEGF and intercellular adhesion molecule 1 (ICAM‐1), thereby substantially reducing vascular leakage and the CNV area [44]. These findings reported herein also demonstrate that significantly higher amounts of HIF1α are produced by ARPE‐19 cells under hypoxia conditions compared to under normoxia conditions. Therefore, these results indicate that HIF1α derived from RPE cells could be a possible stimulator of CNV progression by inducing the transcription of VEGF and ICAM‐1. Indeed, such an HIF1a linked stimulation of the angiogenesis was also recognized as a pivotal mechanism that is associated with the molecular pathogenesis of diabetic retinopathy [49-51]. Furthermore, HIF1α also promotes the TGF‐β2‐induced EMT of human lens epithelial cells [48] as well as that for ARPE‐19 cells [49]. As another important biological aspect of HIFs, it was quite interestingly revealed that HIFs regulate both mitochondrial respiration and mitochondrial oxidative stress, and conversely, mitochondrial metabolism, respiration and oxidative stress also could regulate the activation of HIFs [34]. In fact, Shu et al. reported that the inhibition of PGC1α, a master regulator of mitochondrial function, induced the disruption of the mitochondrial functions and the stimulation of an EMT response within human RPE cells [2]. In their subsequent study, they also demonstrated that TGF-β2 stimulated EMT in RPE, caused the significant down-regulation of PGC1α, and mitochondrial dysfunctions as well as a metabolic shift towards reduced OXPHOS and increased glycolysis [16]. In the current study, although we also found that TGF-b2 and hypoxia synergistically and differently induced EMT of the HRPE cells considering the fluctuation in the mRNA expression of HIF1a and the mitochondrial metabolism described above, those fluctuations were also different between 2D and 3D cultures, and such diversity between them may be caused by possible up-stream regulators requiring the generation of the 3D spheroids, as was recently determined using 3T3-L1 cells [33].”.

Round 2
Reviewer 1 Report
In this revised version, the authors have addressed the reviewers’ comments in the text and considerably improved the figures. Specifically, they have color-coded the graphs for the different types of experiments, and increased the resolution of the images.